# Development of Superhydrophobic Reduced Graphene Oxide (rGO) for Potential Applications in Advanced Materials

**DOI:** 10.3390/nano15050363

**Published:** 2025-02-27

**Authors:** Enoch Adotey, Aliya Kurbanova, Aigerim Ospanova, Aida Ardakkyzy, Zhexenbek Toktarbay, Nazerke Kydyrbay, Mergen Zhazitov, Nurxat Nuraje, Olzat Toktarbaiuly

**Affiliations:** 1Renewable Energy Laboratory, National Laboratory Astana (NLA), Nazarbayev University, Kabanbay Batyr 53, Astana 010000, Kazakhstan; enoch.adotey@nu.edu.kz (E.A.); aliya.kurbanova@nu.edu.kz (A.K.); a.ospanova@nu.edu.kz (A.O.); aida.ardakkyzy@nu.edu.kz (A.A.); nazerke.kydyrbay@nu.edu.kz (N.K.); mergen.zhazitov@nu.edu.kz (M.Z.); nurxat.nuraje@nu.edu.kz (N.N.); 2Department of Chemistry, Faculty of Natural Sciences and Geography, Abai Kazakh National Pedagogical University, 13 Dostyk Ave., Almaty 050010, Kazakhstan; zhexenbek.toktarbay@gmail.com; 3Department of Chemical & Materials Engineering, School of Engineering & Digital Science, Nazarbayev University, Astana 010000, Kazakhstan

**Keywords:** graphene oxide, nanoparticles, reduced graphene oxide, superhydrophobic, hydrophobicity

## Abstract

Reduced graphene oxide (rGO) was synthesized by chemically reducing graphene oxide (GO) using a reducing agent. The product, rGO, showed excellent hydrophobicity, as indicated by its high-water contact angle, which was greater than 150°. Characterizations using Fourier-transform infrared (FTIR) spectroscopy, Raman spectroscopy, and X-ray diffraction (XRD) were used to analyze the composition and structural differences between GO and the superhydrophobic rGO material. Scanning electron microscopy (SEM) showed that GO particles exhibited a plate-like morphology with layers of stacked plates, while rGO displayed fewer stacks that show a more separated structure of layers. The increasing demand for superhydrophobic materials in advanced materials industries, due to their potential to enhance performance, durability, and safety, makes rGO a promising candidate for use in composite materials.

## 1. Introduction

Graphene consists of a single layer of carbon atoms arranged in a hexagonal pattern; it has garnered significant interest due to its remarkable electrical, mechanical, and thermal properties. Graphene oxide (GO), an oxidized form of graphene, has been widely explored for its potential applications such as electronics, energy storage, and composite materials [1,2]. The hydrophilic nature of GO, attributed to its functional groups like epoxides, hydroxyls, and carboxyls, facilitates its dispersion in water-based solutions [3]. However, these groups disrupt the sp^2^ hybridization of the carbon lattice, significantly diminishing graphene’s electrical conductivity and intrinsic properties.

To overcome these drawbacks, scientists have worked on reducing GO to form reduced graphene oxide (rGO), which helps restore graphene’s original characteristics [4,5,6]. The hydrophobic nature of rGO is a key factor influencing its suitability for applications like coatings, membranes, and composites, where water resistance and chemical durability are essential [7]. Hydrophobic materials play a crucial role in preventing moisture absorption, enhancing durability, and ensuring optimal performance in humid or wet environments. The extent of reduction, the type and density of functional groups present, and the overall structure of rGO are critical in determining its hydrophobic properties.

Various approaches have been employed to reduce GO, including chemical, thermal, and electrochemical techniques [8,9]. Among these, chemical reduction has gained significant attention, utilizing reducing agents such as hydrazine, sodium borohydride, and ascorbic acid during synthesis. These substances play a crucial role in eliminating oxygen-containing functional groups from GO, partially restoring the sp^2^ carbon network, thereby enhancing both electrical conductivity and hydrophobicity. Selecting an appropriate reducing agent is essential, as it must effectively minimize the oxygen content in GO to achieve a high degree of reduction and improved hydrophobic characteristics.

Another widely used method for converting GO to rGO is thermal reduction, where GO is exposed to high temperatures in a controlled inert or reducing atmosphere [10]. This process eliminates oxygen functional groups while simultaneously repairing structural defects within the carbon lattice. Although thermal reduction can yield highly reduced graphene, maintaining the precise temperature and atmospheric conditions is necessary to prevent damage to the carbon framework.

Electrochemical reduction is another efficient approach that allows for precise control of the reduction process by applying an electrical potential to GO dispersed in an electrolyte [11]. This technique enables the selective removal of specific functional groups and can be performed at room temperature, making it a more energy-efficient alternative to thermal reduction. Furthermore, electrochemical reduction can be easily scaled up, offering a viable route for large-scale rGO production tailored to specific applications [12,13].

A material’s hydrophobicity is commonly assessed by measuring the contact angle of a water droplet on its surface. A contact angle exceeding 90° classifies the surface as hydrophobic, while values above 150° indicate superhydrophobic properties. Research on hydrophobic coatings and materials has expanded considerably, leading to their widespread use in numerous applications [14,15,16,17].

The hydrophobic behavior of rGO results from the removal of polar oxygen-containing functional groups and the restoration of the non-polar sp^2^ carbon structure. Optimizing the GO reduction process enables the production of rGO with a high contact angle, making it suitable for advanced technological applications. The enhanced hydrophobicity of reduced graphene oxide (rGO) coatings can be achieved through various methods. Zulkharnay et al. demonstrated that electrostatic spray deposition (ESD), followed by thermal annealing under an Ar/H_2_ atmosphere resulted in rGO films with water contact angles exceeding 127°, which was attributed to the removal of oxygen-containing functional groups, which reduced the surface energy and increased the water repellency [17].

In coatings and membranes, hydrophobicity is essential for water repellency, anti-fouling properties, and chemical resistance. Highly hydrophobic rGO can contribute to the development of self-cleaning surfaces, anti-corrosion layers, and protective coatings for electronic devices. In composite materials, its hydrophobic nature enhances dispersion and interfacial bonding with hydrophobic polymers, ultimately improving mechanical strength and durability.

This study focused on optimizing the reduction process of GO to enhance the hydrophobic characteristics of rGO, making it more suitable for use in advanced material technologies. The investigation examined how factors like temperature and the concentration of reducing agents influence the hydrophobic properties of rGO. The findings from this research will provide important insights for the development of superhydrophobic rGO with specialized properties, enabling its application in high-performance materials such as coatings, membranes, and composites.

## 2. Materials and Methods

A modified Hummer’s method was employed to synthesize GO [17]. The key chemicals utilized in this process were primarily obtained from Sigma-Aldrich (St. Louis, MO, USA), including graphite flakes (99.99% purity, natural), sodium nitrate (NaNO_3_, ≥99%), concentrated sulfuric acid (H_2_SO_4_, 95–98%), and potassium permanganate (KMnO_4_, ≥99%). Additionally, deionized (DI) water, hydrogen peroxide (H_2_O_2_, 30%), and hydrochloric acid (HCl, 37%) were used during the oxidation, washing, and purification stages.

### 2.1. Synthesis of GO and rGO

A modified version of Hummer’s method was applied for GO synthesis and its chemical reaction is provided in Equation (1) [18]. Initially, 0.5 g of NaNO_3_ was dissolved in 23 mL of concentrated H_2_SO_4_. Graphite flakes were then added to the well-mixed NaNO_3_-H_2_SO_4_ solution in an ice bath while stirring with a mechanical stirrer. After 20 min, 3 g of KMnO_4_ was gradually incorporated in three separate 1 g portions at 15 min intervals. Upon the first addition, the solution’s temperature began to rise, and its color shifted to dark green due to the formation of the oxidizing agent MnO_4_^2−^. The reaction mixture was continuously stirred for 24 h, at the end of which, the solution turned dark purple, signaling the completion of oxidation. Following this 24 h period, approximately 50 mL of DI water was slowly introduced while keeping the system in an ice bath to control the temperature. After 20 min, an additional 125 mL of DI water was carefully added while stirring vigorously for 25 min. Then, 1 mL of H_2_O_2_ was introduced at 5 min intervals (three times, totaling 3 mL), followed by continuous stirring for another 25 min. The mixture underwent three washes with 10% HCl and two rinses with water. To exfoliate the graphene oxide, the solution was ultrasonicated for 2 h, and then centrifuged and dried at 120 °C for 8 h. Finally, thermal annealing of GO at 350 °C for 2 h resulted in the formation of graphene oxide.4C + 2KMnO_4_ +2NaNO_3_ +2H_2_SO_4_ → 4GO + 2MnO_2_ + K_2_SO_4_ + Na_2_SO_4_ + 2NO_2_ + 2H_2_O(1)

### 2.2. Reduction of Graphene Oxide

Approximately 500 mg of the synthesized graphene oxide was dispersed in 10 mL of deionized water and subjected to sonication for 10 min. The mixture was then gradually heated to 40 °C while being stirred continuously. Once the desired temperature was reached, 1 mL of hydrazine hydrate was introduced, and the solution was stirred for an additional hour. After the reaction was complete, the final product was collected through centrifugation and then dried overnight in an oven at 60 °C.

### 2.3. Stability of the rGO in Different Solvents

Different solvents were used to disperse the rGO particles at a concentration of 1 mg/mL, and their stability was observed after 1 h and again after 24 h. The organic solvents selected for this study included ethanol, acetone, cyclohexane, toluene, dichloroethane, and dimethylbenzene.

### 2.4. Material Characterization

The morphology of the samples was examined using a ZEISS Crossbeam 540 scanning electron microscope (Oberkochen, Germany). High-resolution SEM images were captured at an accelerating voltage of 15 kV and a 5 nm spot size. To reduce charging effects, a thin gold coating was applied to the samples before analysis. Raman spectroscopy was conducted with a LabRAM (Horiba, Kyoto, Japan) Raman spectrometer (France SAS, Paris, France), utilizing a 473 nm blue laser with a power of 10 mW and a 1 µm spot size to ensure high spatial resolution. Spectral measurements were taken between 1000 and 2500 cm^−1^ and under ambient conditions. Water contact angle (OCA) measurements were carried out using a Dataphysics OCA 15 Pro goniometer (Filderstadt, Germany). A 5 µL water droplet was placed on the sample’s surface, and static contact angles were determined by analyzing its shape. The final value was reported as the average of three measurements. Infrared spectroscopy was performed with a Nicolet iS10 FTIR spectrometer (Thermo Fisher Scientific, Waltham, MA, USA), recording spectra in the range of 4000 cm^−1^ to 400 cm^−1^ with a spectral resolution of 4 cm^−1^. Each spectrum was obtained by averaging 64 scans, using air as the background reference for precise data collection. X-ray photoelectron spectroscopy (XPS) was conducted using a Thermo Scientific NEXSA XPS system (Waltham, MA, USA) with an Al Kα X-ray source (1486.6 eV). Survey scans were recorded at a pass energy of 200 eV, while high-resolution scans were acquired at 50 eV to improve energy resolution. The binding energy scale was calibrated using the C 1s peak at 284.8 eV, and spectra were analyzed using the Shirley background subtraction method, with peak fitting performed via a Gaussian–Lorentzian (GL) function. X-ray diffraction (XRD) patterns were collected with a Rigaku SmartLab^®^ X-ray diffraction (Tokyo, Japan) system utilizing Cu-Kα radiation (λ = 1.5406 Å). Data were recorded over an angular range of 10° to 60° at a scanning speed of 5° min^−1^, allowing for the evaluation of the crystalline structure and phase composition of the samples.

## 3. Result and Discussion

### 3.1. Characterization

The TEM and SEM images (Figure 1a,b) illustrate the morphology of GO. According to previous studies [19], the chemical transformation of graphite into graphene introduces defects and holes in the carbon lattice. These structural imperfections likely result from the elimination of oxygen-containing functional groups during the reduction process. Additionally, nanocomposites can form in proximity to these defects and become embedded within the graphene sheets [20]. As observed in Figure 1b, the oxidation of graphite led to a decrease in the number of layers, a reduction in crystallinity, and increased amorphization. The TEM analysis confirmed that the synthesized GO consisted of a single-layer structure with noticeable defect sites [21].

A detailed comparison between the SEM images of GO and rGO was conducted, as shown in Figure 2. The morphology of GO appeared as layered nanoplates adhering to one another. The graphene oxide image (Figure 2a) reveals stacked structures, indicating that exfoliation of graphite had taken place. The SEM analysis further demonstrated that GO particles contain high-volume pores forming a “mesh-like” structure, which could serve as a macroscopic arrangement of aligned graphene sheets. This structure has many potential applications, including strong fiber production, functional textiles, field-emission displays, and energy storage [22]. However, since the objective of this study was to obtain a hydrophobic material, GO was reduced using hydrazine hydrate before further processing. The SEM image in Figure 2b displays noticeable wrinkles and crumples on the rGO surface. Wrinkling is a typical characteristic of 2D materials and is primarily caused by the negative thermal expansion of rGO [23,24,25,26,27]. Additionally, the crumpled structures resemble “paper-ball-like” formations, which resulted from isotropic compression and the thermal reduction process of the nanomaterial [25].

X-ray diffraction (XRD) was utilized to examine the crystal phase and measure the interlayer spacing. Figure 1c presents the spectra for graphite, GO, and rGO, which align with previously reported findings in the literature [19,28,29,30,31,32,33]. Graphite displayed a sharp peak at 2θ = 26.55°, signifying a well-ordered layered structure with a d-spacing of 0.335 nm along the (002) orientation. In the case of GO, the peak shifted to 11.5°, confirming the complete oxidation of graphite into GO [32,33]. This shift indicates an increased interlayer distance, which results from the insertion of oxygen-containing functional groups—such as epoxy, hydroxyl, carbonyl, and carboxyl groups—into the carbon basal plane during chemical oxidation [34,35,36]. Consequently, the separation between the carbon layers expands. After thermal reduction, which effectively removes most oxygen functional groups, rGO exhibited a peak at 2θ = 24.62°, suggesting a partial restoration of graphene’s π-conjugated structure [34,37,38]. Unlike the sharp and well-defined (002) peak of graphite, rGO displayed a broader (002) peak, indicating a more disordered crystalline arrangement rather than a highly ordered structure [39,40,41,42]. Additionally, the reduction in d-spacing confirmed the efficient removal of oxygen-containing groups [43,44].

The Raman spectroscopy results for GO and rGO are depicted in Figure 1d. The GO spectrum revealed a prominent D band, which corresponds to the transition from sp^2^ to sp^3^ hybridized carbon, while the G band is associated with the in-plane stretching vibrations of sp^2^ carbon atoms [45]. The D band provides insight into the structural defects in graphene, whereas the 2D band indicates the stacking order and the number of layers [46,47]. Some researchers have suggested that the D band represents breathing modes or ring vibrations of K-point phonons with A1g symmetry [48]. The G band in the GO sample at 1595 cm^−1^ with an intensity of 2979 characterizes carbon materials. The D band at 1367cm^−1^ with an intensity of 2685 indicates defects and partial disorder introduced during oxidation in the carbon materials [49]. The peak placement was observed, which indicates an amorphous structure [23]. In the case of rGO, peaks at D (1367 cm^−1^, 882) and G (1589 cm^−1^, 944) appeared [50].

Fourier-transform infrared (FTIR) spectroscopy was used to identify the functional groups in GO and validate its oxidation process, as shown in the spectrum in Figure 3. The key absorption bands included a broad peak between 3590 and 2500 cm^−1^, associated with carboxyl O-H stretching; an O-H stretching peak around 3375 cm^−1^, attributed to adsorbed water and alcohol groups; a peak at 1616 cm^−1^ corresponding to C=C stretching in unoxidized graphitic regions; a C=O stretching peak at 1713 cm^−1^ from carboxyl groups; a 1221 cm^−1^ peak for C-OH stretching in alcohols; and a peak at 1032 cm^−1^, signifying C-O-C stretching. After reduction with hydrazine hydrate, the rGO’s FTIR spectrum exhibited a significant decrease in intensity and the peaks related to oxygen-containing groups disappeared, confirming successful reduction while suggesting that some functional groups remained [26].

XPS characterization of the GO and rGO particles was conducted. The survey spectra of GO and rGO displayed in Figure 4a,b show that the major constituents were carbon and oxygen. Figure 4c shows the C 1s XPS spectra for GO. After deconvolution, the C 1s spectrum of GO revealed a peak at 284.5 eV, which corresponds to the sp2 C–C bonds carbon peak. The binding energy of the oxygen functionalities were present as C–O–C, C=O, and O=C–OH, which were assigned to 286.5, 287.5, and 289.3, respectively. A peak appeared at 289.9 eV which was suspected to be a COOH bond [51]. The deconvoluted peaks for rGO are shown in Figure 4d. An intense carbon band at 284.4 eV in the rGO spectrum was visible and corresponds to graphitic carbon. A peak at 286.3 eV was present and was assigned as C-OH (epoxy/hydroxy), while a smaller peak at 288.6 eV indicated the presence of oxygen-containing O–C=O bonds in the rGO structure [50,51,52,53].

The stability of rGO in various organic solvents was evaluated, and the results are presented in Figure 5. This analysis aimed to identify the most effective solvent for dispersing rGO, a critical factor for its integration into coatings, composites, and electronic applications. The ability of rGO to remain well dispersed directly impacts its processability and overall performance in different applications. Among the tested solvents, ethanol and toluene demonstrated excellent dispersion properties. When rGO was dispersed at a concentration of 1 mg/mL in these solvents and allowed to stand for 60 min, only a minimal amount of particle sedimentation was observed in both cases. Notably, after 24 h, no additional precipitation occurred in ethanol, indicating its strong stabilizing effect on rGO dispersions over time. The role of hydrazine as both a reducing agent and a surface modifier was also evident in the enhanced dispersion stability. By facilitating the removal of oxygen functional groups and converting sp^3^ covalent bonds into sp^2^ hybridized carbon, hydrazine contributed to the enhanced stability of rGO in solution. The higher sp^2^ carbon content strengthened π-π interactions with ethanol, reinforcing its effectiveness as a dispersion medium for rGO.

### 3.2. Hydrophilic and Hydrophobic Properties

The water contact angle measurement was used to examine the wetting behavior of GO and its derivatives, with the results illustrated in Figure 6. A material is classified as hydrophilic if its contact angle is below 90°, hydrophobic if the angle exceeds 90°, and superhydrophobic if it exceeds 150°. As shown in Figure 6, GO exhibits hydrophilic properties, which can be attributed to the abundance of functional groups such as hydroxyl, carbonyl, and carboxyl groups. However, once GO underwent reduction, most of these functional groups were eliminated, transforming rGO into a superhydrophobic material with a contact angle of 155°. The water contact angle was determined using a contact angle goniometer (OCA 15 Pro). Small droplets of deionized water (approximately 3–5 μL) were precisely placed onto the sample surface using a micropipette, and the angle formed at the interface between the droplet and the surface was recorded. Multiple measurements were conducted at different locations to ensure the reliability and accuracy of the results, and an average contact angle was calculated for each sample.

Research has indicated that wet road conditions contribute to frequent highway accidents, mainly due to hydroplaning and reduced skid resistance. These two factors are major concerns for highway engineers and researchers. The presence of water on road surfaces significantly increases the risk of these hazards. Applying superhydrophobic coatings to pavement surfaces is an innovative approach to mitigating these risks. Such coatings function by reducing adhesion, thereby enhancing water repellency on the road surface. Given the excellent superhydrophobic characteristics of rGO, it has potential applications in surface coatings. Methods such as the layer-by-layer coating technique can be employed, where a base coat ensures adhesion, followed by a top layer of superhydrophobic rGO to effectively repel water. Integrating rGO into advanced materials for road infrastructure presents a promising avenue for future research.

## 4. Conclusions

GO was synthesized, and hydrazine hydrate was employed as a reducing agent to obtain rGO. The resulting rGO exhibited a high water contact angle, confirming its strong hydrophobic properties. Various characterization techniques, including FTIR spectroscopy, Raman spectroscopy, and XRD, were utilized to analyze the structural and elemental composition of both the GO and superhydrophobic rGO samples. SEM imaging revealed that the GO particles appeared as layered plates, while rGO displayed a structure with fewer stacked layers. With the increasing demand for superhydrophobic materials in industrial applications due to their potential to improve performance, longevity, and safety, rGO has emerged as a promising candidate for use in composite materials alongside other substances.

## Figures and Tables

**Figure 1 nanomaterials-15-00363-f001:**
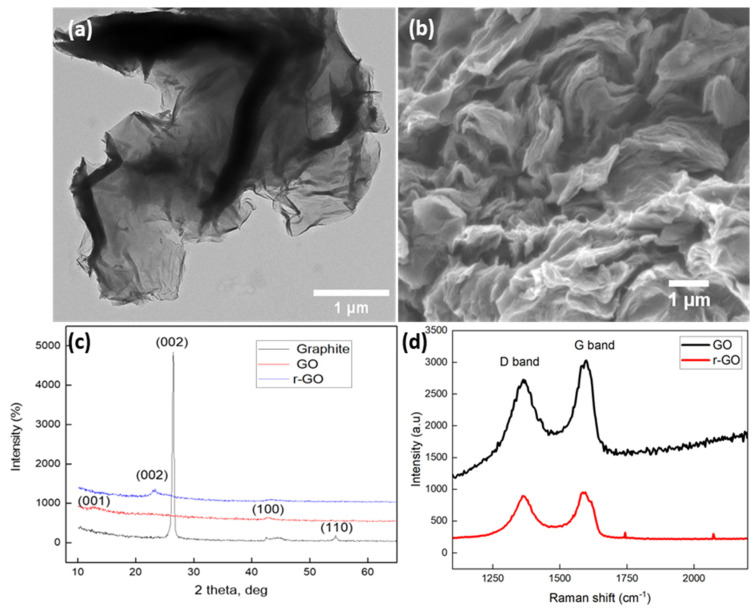
(**a**) TEM image of GO; (**b**) SEM image of GO; (**c**) XRD patterns of graphite, GO, and rGO; (**d**) Raman spectra of GO and rGO.

**Figure 2 nanomaterials-15-00363-f002:**
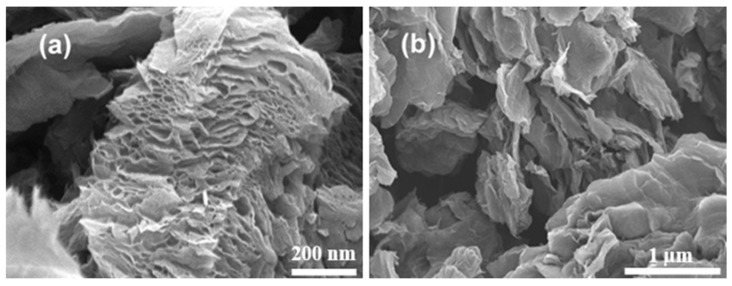
SEM topography images of (**a**) GO and (**b**) rGO.

**Figure 3 nanomaterials-15-00363-f003:**
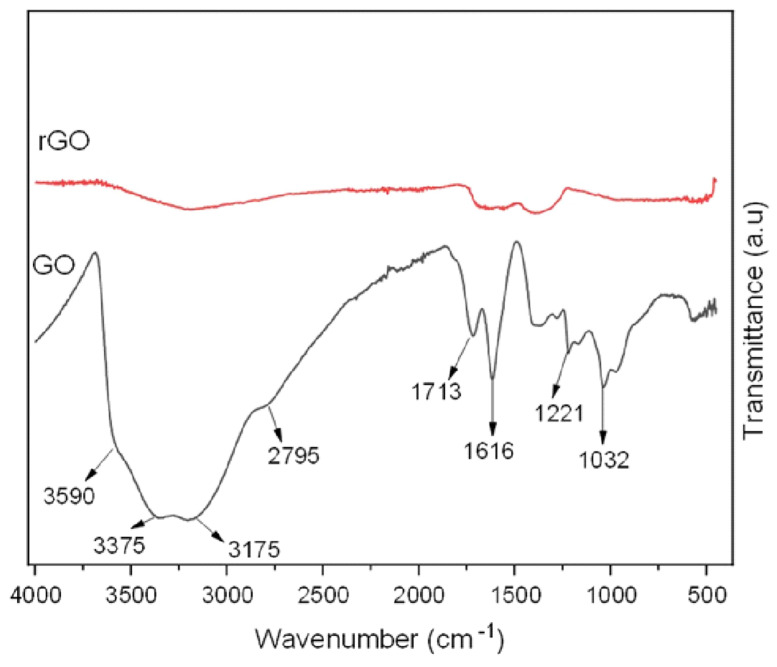
FTIR spectra of GO and rGO.

**Figure 4 nanomaterials-15-00363-f004:**
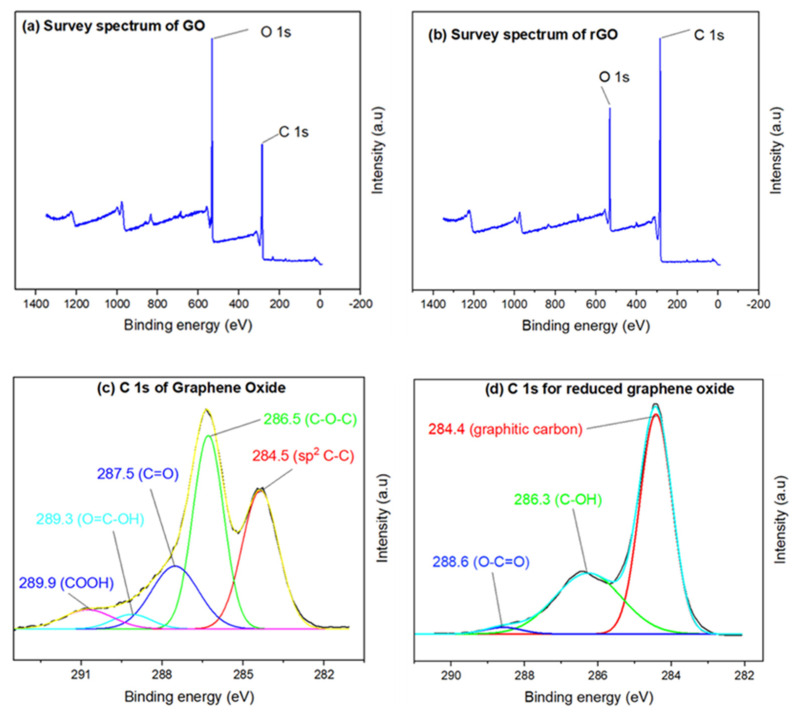
XPS survey spectra of (**a**) GO and (**b**) rGO: C 1s spectrum for (**c**) GO and (**d**) rGO.

**Figure 5 nanomaterials-15-00363-f005:**
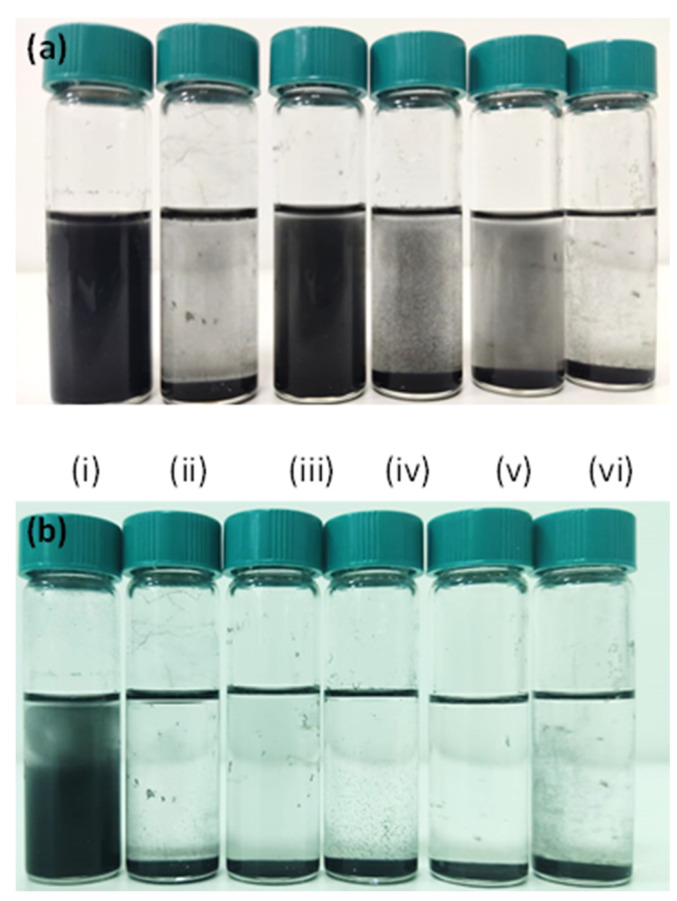
Stability of rGO in different solvents, with (**a**) representing the initial dispersion and (**b**) showing the sedimentation and aggregation after 24 h. From left to right: ethanol, toluene, methanol, dichloroethane, acetone, and hexane.

**Figure 6 nanomaterials-15-00363-f006:**
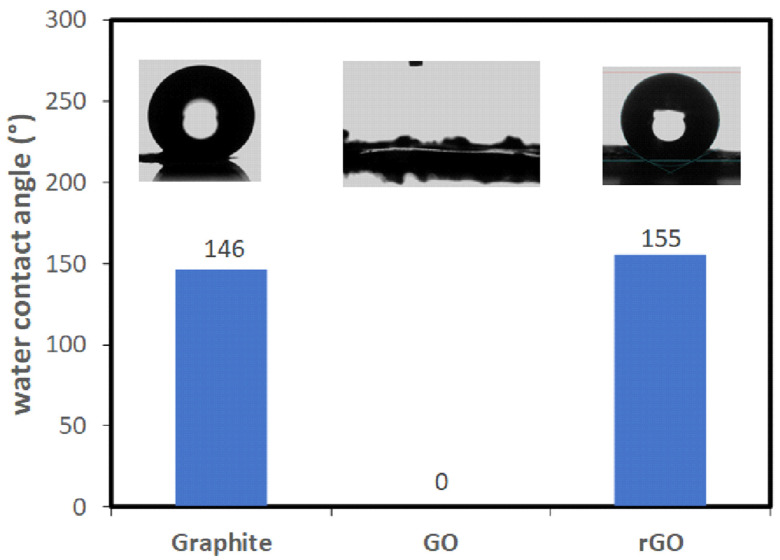
Water contact angles of graphite, GO, and rGO.

## Data Availability

The data supporting the conclusions of this article will be made available by the corresponding author upon request.

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
