# Peer review of "Development of Superhydrophobic Reduced Graphene Oxide (rGO) for Potential Applications in Advanced Materials"

_nanomaterials, 2025, doi:10.3390/nano15050363_

Round 1

Reviewer 1 Report

Comments and Suggestions for Authors

This manuscript provided a method to prepare graphene oxide materials, and their hydrophobicity was tested. However, this seemed as an unfinished work. Therefore, it cannot be recommended based on the present manuscript.

1.     Please check Table 1, what was the meaning of it?

2.     As showed in title “…in Advanced Materials”, what kinds of advanced materials? Any examples?

3.     How to make the coatings of materials to give an enhanced hydrophobicity?

4.     What was the meaning of “SH coating” in Line 248?

5.     What about the process details of the water contact angle measurement?

6.     In Part 2.4, “…angular range of 5-40°and a scanning rate…”, however, in Fig. 1C? Please check?

7.     The purpose of “Stability of the rGO in different solvents” should be explained in the main content.

Reviewer 2 Report

Comments and Suggestions for Authors

Reviewer report on manuscript nanomaterials-3472564

Enoch Adotey et al.Optimizing Graphene Oxide Reduction for Improved Hydrophobicity in Advanced Materials

In present work, reduced graphene oxide (rGO) was synthesized by chemically reducing graphene oxide (GO) using a reducing agent. The product, rGO, showed good hydrophobicity, as indicated by its high-water contact angle, greater than 150°. Characterizations, such as Fourier-transform infrared spectroscopy (FTIR), Raman spectroscopy, and X-ray diffraction (XRD), were used to analyze the composition and structural differences between GO and the superhydrophobic rGO materials. Scanning electron microscopy (SEM) showed that GO particles exhibited a plate-like morphology with layers of stacked plates, while rGO displayed fewer stacks, that show a more separated structure of layers.

The manuscript can be accepted only after major revision.

Below, I point out several questions to help the authors improve the manuscript before publication.

Questions/comments:

1.      The manuscript needs extensive English revision!

2.      The manuscript is very carelessly formatted and requires extensive editing!

3.      The biggest part of the references used is not up to date.

4.      More details to the section “2.4.Material Characterization” should be added, including information about X-ray photoelectron, Raman and FT-IR spectra acquisition, and SEM, TEM images and XRD patterns collecting.

5.      The excitation energy, energy resolution and pass energy values during X-ray photoelectron spectra acquisition should be written. The procedures for background subtraction and spectra fitting should be well described.

6.      Survey photoelectron spectra for all samples should be provided also. Without them, it’s not possible to make such conclusion (Page 6): “The XPS spectrum of GO/rGO did not contain any elements other than C and O, indicating the absence of impurities.

7.      The information about the XPS spectra calibration should be provided.

8.      The assignment of the peaks in the Raman spectra (Figure 1d) are not well justified. The up-to-date references (2024) should be added. I recommend Authors using the publication [J. Alloys Compd. 2020, 849, 156699], [Carbon 2022, 194, 52], and references there.

9.      The peaks identification in the FTIR spectra is not justified. There are no up-to-date references. I recommend using the publication [Carbon, 2022, 196, 264-279].

10.  It is well-known that sp2 component of the X-ray photoelectron spectra (Figure 4) has an asymmetric shape from the side of higher binding energies, which is described by the Doniach-Sunjic function [J. Environ. Chem. Eng. 2022, 10(3), 107873]. Why didn’t authors implement the asymmetry in their deconvolution procedure?

11.  The C1s XPS fitting (Figures 4a, 4b) is not very good justified. The carbonyl component is missed. The sp2 component should have the binding energy of 284.8 eV. There are not up-to-date references (2024) for choice of components. The references to up-to date 2024 investigations in this field should be added. For the identification of C-O moieties, I can recommend using publications [J. Alloys Compd. 2020, 849, 156699], and references there.

12.  The piece of text on pages 2-3 should be removed. It repeats twice, namely “The concept of Hummer's method was modified and used to synthesize GO [17]. First, 0.5 g NaNO3 was dissolved in 23 mL of concentrated H2SO4. The graphite flakes were then added to a well-mixed solution of NaNO3 and H2SO4 while it was kept in an ice  bath and mixed with a mechanical stirrer. After 20 minutes, 3g of KMnO4 was added by splitting it into three portions of 1g each at a 15-minute interval. After adding the first portion, the temperature of the solution gradually increased, and the color of the solution changed to dark green due to the creation of the oxidizing agent MnO3. The solution was left stirring for 24 hours and by the end of this period, it became dark purple indicating the end of oxidation. After 24 hours, about 50ml DI water was slowly added to the system, using an ice bath to reduce the temperature. After 20 minutes, another 125ml of DI water was slowly added while vigorously stirring for 25 minutes. Then, at 5-minute intervals, 1 ml of H2O2 was added (3 times, in a total of 3 ml) and stirred continuously for the next 25 minutes. The resulting solution was rinsed three times with 10% HCl and twice with water. Finally, the solution was ultrasonicated for 2 hours to exfoliate the graphene oxide. It was then centrifuged and dried at 120 °C for 8 hours. Reduced graphene oxide was obtained by thermal annealing of GO at 350°C for 2h.

13.  The caption of Figure 3 “Figure 3. SEM topography images of (a) GO and (b) rGO.” should be replaced. There are FT-IR spectra of GO and rGO.

14.  Table 1 (Page 8) should be removed. It is just a part of the journal template.

15.  The reference [36] should be corrected. Part of the required information is missed.

16.  The reference [40] „Moustafa, S. H. S., Amorphous Carbon (aC: H): Atomic Bonding Structure, Electrical and Optical Properties, PhD thesis booklet 352, (1993)“ should be removed. It is wrong and doesn’t provide necessary information.

17.  The reference [41] should be corrected. The part of the required information is missed.

18.  Statement about “Data availability” should be re-written in correct way.

All subscript and superscript characters are skipped in the text. It should be corrected.

Comments on the Quality of English Language

The manuscript needs extensive English revision!

Round 2

Reviewer 1 Report

Comments and Suggestions for Authors

In this manuscript, how to make the coatings of materials to give an enhanced hydrophobicity was not showed in this study. It was a potential application? So the title was not so proper. Please consider.

Author Response

Comment 1: In this manuscript, how to make the coatings of materials to give an enhanced hydrophobicity was not showed in this study. It was a potential application? So the title was not so proper. Please consider.

Response 1: Thank you for pointing this out. We agree with your comment and have revised the title as follows:

"Development of Superhydrophobic Reduced Graphene Oxide (rGO) for Potential Applications"

Reviewer 2 Report

Comments and Suggestions for Authors

Manuscript can be accepted in the revised form.

Author Response

Thanks for accepting the revised version. We have made changes to the title: 

"Development of Superhydrophobic Reduced Graphene Oxide (rGO) for Potential Applications."